# IL-22-Activated *MUC13* Impacts on Colonic Barrier Function through JAK1/STAT3, SNAI1/ZEB1 and ROCK2/MAPK Signaling

**DOI:** 10.3390/cells12091224

**Published:** 2023-04-23

**Authors:** Tom Breugelmans, Wout Arras, Baptiste Oosterlinck, Aranzazu Jauregui-Amezaga, Michaël Somers, Bart Cuypers, Kris Laukens, Joris G. De Man, Heiko U. De Schepper, Benedicte Y. De Winter, Annemieke Smet

**Affiliations:** 1Laboratory of Experimental Medicine and Pediatrics, Faculty of Medicine and Health Sciences, University of Antwerp, 2610 Antwerp, Belgium; tom.breugelmans2@uantwerpen.be (T.B.);; 2Infla-Med Research Consortium of Excellence, University of Antwerp, 2610 Antwerp, Belgium; 3Department of Gastroenterology and Hepatology, University Hospital of Antwerp, 2650 Antwerp, Belgium; 4Department of Computer Science, Adrem Data Lab, University of Antwerp, 2610 Antwerp, Belgium

**Keywords:** transmembrane mucin, tight junction, intracellular signaling, inflammation

## Abstract

Overexpression of the transmembrane mucin MUC13, as seen in inflammatory bowel diseases (IBD), could potentially impact barrier function. This study aimed to explore how inflammation-induced MUC13 disrupts epithelial barrier integrity by affecting junctional protein expression in IBD, thereby also considering the involvement of MUC1. RNA sequencing and permeability assays were performed using LS513 cells transfected with *MUC1* and *MUC13* siRNA and subsequently stimulated with IL-22. In vivo intestinal permeability and MUC13-related signaling pathways affecting barrier function were investigated in acute and chronic DSS-induced colitis wildtype and *Muc13*^−/−^ mice. Finally, the expression of *MUC13*, its regulators and other barrier mediators were studied in IBD and control patients. Mucin knockdown in intestinal epithelial cells affected gene expression of several barrier mediators in the presence/absence of inflammation. IL-22-induced *MUC13* expression impacted barrier function by modulating the JAK1/STAT3, SNAI1/ZEB1 and ROCK2/MAPK signaling pathways, with a cooperating role for MUC1. In response to DSS, MUC13 was protective during the acute phase whereas it caused more harm upon chronic colitis. The pathways accounting for the MUC13-mediated barrier dysfunction were also altered upon inflammation in IBD patients. These novel findings indicate an active role for aberrant MUC13 signaling inducing intestinal barrier dysfunction upon inflammation with MUC1 as collaborating partner.

## 1. Introduction

Inflammatory bowel diseases (IBD; Crohn’s disease [CD] and ulcerative colitis [UC]), characterized by chronic relapsing inflammation of the intestines, are major health problems in developed countries [1]. There is emerging evidence that defects in the intestinal mucosal barrier function could be significant contributors to the IBD pathophysiology [2,3]. The intestinal mucosal barrier comprises a thick mucus layer, a monolayer of intestinal epithelial cells (IECs) and the lamina propria hosting innate and adaptive immune cells [4]. IECs form a continuous lining that functions as a physical barrier by regulating paracellular permeability and limiting access of microbes and toxins to the underlying tissues, and is protected by a mucus layer [5]. Secreted and transmembrane mucins (MUC) are the main gatekeepers of the mucus barrier, comprising a large family of high molecular weight glycoproteins that are characterized by heavily glycosylated domains rich in proline, threonine and serine (i.e., PTS domains) [4]. Whereas the secreted MUC2 mucin—produced by goblet cells—is the major constituent of the mucus layer, cell-surface mucins (MUC1, MUC3, MUC4, MUC13 and MUC17) present a dense network on the apical side of IECs forming the glycocalyx [4]. Besides providing a barrier to potential pathogens, transmembrane mucins also participate in intracellular signal transduction and play an important role in the maintenance of IEC homeostasis [4]. Aberrant mucin expression, as characterized by a reduction in MUC2 secretion and overexpression of transmembrane mucins, has been described in IBD [6,7,8,9]. MUC2 expression is essential in suppressing the inflammatory response by blocking access of bacteria to the underlying epithelium, and the absence of *Muc2* has been shown to predispose mice to the development of spontaneous colitis [9]. MUC13 is one of the most abundant transmembrane mucins in the intestinal tract, whereas MUC1, a predominant gastric cell-surface mucin, is expressed at low levels on the luminal side of the colonic epithelial cells. However, the expression of both mucins is significantly upregulated in the inflamed mucosa of IBD patients [7]. Such deviant overexpression can affect barrier integrity by modulating signaling pathways and junctional protein function and could be responsible for the progression towards disease [4,10]. Indeed, we have recently described strong positive associations between aberrant *Muc1* and *Muc13* expression, increased intestinal barrier permeability and the altered expression levels of junctional and polarity proteins during acute and chronic colitis in dextran sodium sulphate (DSS)-treated mice [7]. Furthermore, MUC1 and MUC13 have opposing influences on inflammation [11]. More specifically, *Muc1^−^^/−^* mice were resistant to acute DSS-induced colitis, yet were more prone to gastrointestinal infection [12,13,14]. Conversely, *Muc13*^−/−^ mice are more susceptible to acute DSS-induced colitis by promoting epithelial cell apoptosis but deficiency of *Muc13* expression during chronic colitis reduced the risk for tumor formation in azoxymethane (AOM)/DSS-treated mice [8,15]. In addition, the absence of *Muc1* also decreased the risk for colorectal cancer development in mice [16]. The above findings clearly highlight that both MUC1 and MUC13 are dual-faceted proteins providing a protective barrier function but impacting intestinal barrier homeostasis via constitutive activation of signaling pathways upon chronic inflammation. Compared to MUC1, MUC13 is atypical in that it has a short extracellular domain but contains a 69 amino acid cytoplasmic domain with eight serine and two tyrosine residues for potential phosphorylation and a protein kinase C consensus phosphorylation motif, further suggesting that MUC13 plays a key role in IEC homeostasis, regulating apoptosis and proliferation [4,17]. How MUC13 disrupts the epithelial barrier integrity by affecting epithelial cell–cell interactions and cell polarity upon intestinal inflammation in IBD has not yet been studied. Furthermore, interleukin (IL)-22 plays a crucial role in promoting epithelial cell homeostasis in the gastrointestinal tract by regulating barrier function via, amongst others, mucin and tight junction expression [18,19]. However, exuberant production of IL-22 under pathological conditions, such as IBD, may cause epithelial inflammation and the aberrant expression of mucins, such as MUC13 [20]. Here, we performed in-depth molecular analyses to unravel the key *MUC13*-mediated signaling pathways involved in intestinal barrier dysfunction upon IL-22 activation in IBD using in vitro, in vivo and translational approaches. As a strong correlation between *MUC1* and *MUC13* expression has been described in IBD [7,8,21], the link with MUC1 as an important partner in crime was also considered.

## 2. Material and Methods

### 2.1. Cell Lines

LS513 cells (human colorectal carcinoma [ATCC© CRL-2134™]) were grown at 37 °C and 5% CO_2_ in RPMI-1640 medium (Gibco™, Thermo Fisher Scientific, Paisley, Scotland) supplemented with 10% heat-inactivated fetal bovine serum (Gibco™, Thermo Fisher Scientific, Waltham, MA, USA), 100 U/mL penicillin (Gibco™, Thermo Fisher Scientific, Waltham, MA, USA), 100 µg/mL streptomycin (Gibco™, Thermo Fisher Scientific, Waltham, MA, USA) and 2 mM L-glutamine (Gibco™, Thermo Fisher Scientific, Waltham, MA, USA). Medium was replaced every two days.

### 2.2. siRNA Transfection Experiments

LS513 cells were transfected when reaching 70% confluence by adding 75 pmol of Silencer Select *MUC1* (s531542) and/or *MUC13* (s32232) and Negative Control No. 1 (4390843) siRNA (control siRNA, Invitrogen, Thermo Fisher Scientific, Waltham, MA, USA) using Lipofectamine RNAiMAX reagent (13778150, Invitrogen, Thermo Fisher Scientific, Waltham, MA, USA) according to the manufacturer’s instructions. Upon full confluence (after 48 h), the cells were treated with 20–40 ng/mL human recombinant IL-22 (200-22, PeproTech, Thermo Fisher Scientific, Waltham, MA, USA) for 24 h.

### 2.3. Transwell Intestinal Permeability Assays

LS513 (2 × 10^5^) cells were seeded on 12 mm transwells with 0.4 µm pore polyester membrane inserts (3460, Corning, Amsterdam, The Netherlands). Each day, TEER was analyzed using chopstick electrodes on a Millicell^®^ ERS-2 Volt-Ohm Meter (C77976, Merck, Darmstadt, Germany). When reaching 70% confluence, cells were transfected by adding 15 pmol of Silencer Select *MUC13* siRNA using Lipofectamine RNAiMAX reagent on the apical side, which was repeated every other day. When TEER reached a plateau (i.e., after ca. ten days), cells were stimulated with 100 ng/mL IL-22 on their basolateral side for 48 h. Untreated cells and cells transfected with ctrl siRNA were included as controls. Subsequently, intestinal monolayer permeability was analyzed by adding 200 µL of a 4 mg/mL 4 kDa FITC-dextran (46-944, Sigma-Aldrich, Merck, Darmstadt, Germany) solution in Hank’s Balanced Salt Solution (HBSS [1X], Gibco™, Thermo Fisher Scientific, Waltham, MA, USA) on the apical side. 500 µL of HBSS containing no FITC-dextran was added to the basolateral side. After 90 min, samples from the basolateral chamber were collected and aliquots of 100 µL were added in duplicates to a 96-well dark microplate after which the concentration of FITC-dextran was measured by spectrophotofluorometry (Fluoroskan Microplate Fluorometer, Thermo Fisher Scientific, Waltham, USA) at an excitation wavelength of 480 nm and an emission wavelength of 530 nm. The relative FITC-dextran concentration per well was calculated as a percentage of fluorescence relative to a transwell insert containing no cells (No Cell Control).

### 2.4. Generation of Muc13-Deficient Mice

Using CRISPR-Cas9 technology, a 67 bp deletion was induced in exon 4 of the *Muc13* gene in C57BL/6J mice, resulting in a premature stop codon and a knockout by Nonsense Mediated Decay (NMD) of the mRNA. Heterozygous mice were crossed to generate adequate numbers of male knockout (*Muc13*^−/−^) and littermate control (*Muc13*^+/+^, *Muc13*^+/−^) mice for further experiments. Gene deletion was validated by gel electrophoresis and the absence of MUC13 expression in *Muc13*^−/−^ mice was validated by RT-qPCR and immunohistochemistry. All animals were housed in a conventional animal facility with ad libitum access to food and water and a light cycle of 12 h.

### 2.5. Induction of Murine DSS-Induced Colitis and Assessment of Clinical Disease Activity

At least one week before starting the experiments, both knockout and littermate control mice from different litters were co-housed to control for potential differences in gut microbiota composition. Acute and chronic colitis were induced by administering 2% DSS (36–50 kDa, 42867, Sigma-Aldrich, Merck, Darmstadt, Germany) to autoclaved drinking water for 7 days ad libitum with intermediate recovery phases of 7 days with normal drinking water [7]. Administration of 3% DSS for 7 days was used to induce and evaluate acute colitis. Control mice received only autoclaved drinking water. Water levels were checked every day and were refreshed every other day. Every two days, an individual disease activity index (DAI) was calculated by analyzing weight loss (0 ≤ 1%, 1 = 1–5%, 2 = 5–10%, 3 = 10–20%, 4 ≥ 20%), stool consistency (0 = normal, 1 = semi-solid, 2 = loose stools, 4 = diarrhea) and rectal bleeding (0 = no bleeding, 2 = blood visible, 4 = gross bleeding) to obtain a cumulative score of these parameters ranging from 0 (healthy) to 12 (severe colitis). In addition, intestinal inflammation was also monitored in a continuous manner in individual mice by weekly colonoscopy using the Karl Storz Coloview mini-endoscopic system (Karl Storz, Tuttlingen, Germany). Briefly, mice were sedated intraperitoneally (ip) with a mixture of ketamine (60 mg/kg, Ketalar, Pfizer, Brussels, Belgium) and xylazine (6.67 mg/kg, Rompun, Bayer, Leverkusen, Germany) and placed in a prone position. The anal sphincter was lubricated with gel (RMS-endoscopy) to facilitate insertion of the endoscopic sheath (3.5 mm diameter, Karl Storz). Subsequently, the scope was carefully inserted through the anus as far as possible into the colon of the sedated mouse. The murine endoscopic index of colitis severity (MEICS) score was determined during withdrawal of the scope for the following parameters: morphology of the vascular pattern, bowel wall translucency, fibrin attachment, granularity and stool consistency (each ranging from 0 to 3), with a cumulative minimum of 0 (no inflammation) and a maximum of 15 (severe inflammation). At the end of each DSS treatment, a group of mice was sacrificed, the colon resected, feces removed and the weight as well as the length of the colon determined and expressed as the weight/length ratio (mg/cm). Thereafter, different colonic samples were taken from distal to proximal side and stored until further analysis according to a fixed sequence for the following analyses: tissue stored in RNA*later* (Sigma-Aldrich, Merck, Darmstadt, Germany) for evaluating mRNA expression, snap frozen tissue for evaluating protein expression, snap frozen tissue for evaluating MPO activity, and tissue embedded in paraffin for immunohistochemistry. All animal experiments were approved by the Ethical Committee for Animal Testing of the University of Antwerp, Belgium (ECD-file 2017-43).

### 2.6. Quantification of In Vivo Intestinal Permeability in Mice

To assess in vivo intestinal permeability, the FITC-dextran intestinal permeability assay was performed as described previously [7,22]. In brief, mice were orogastrically inoculated 4 h prior to euthanasia with 4 kDa FITC-dextran (44 mg/100 g body weight). Upon euthanasia, blood was collected via cardiac puncture and transferred into SSTII Advance Blood Collection Tubes (BD Vacutainer). After centrifugation (10,000 rpm, 5 min), serum was collected and equally diluted with phosphate-buffered saline (PBS [1X], Gibco™, Thermo Fisher Scientific, Waltham, MA, USA). Subsequently, aliquots of 100 µL were added in duplo to a 96-well microplate and the concentration of FITC measured by spectrophotofluorometry (Fluoroskan Microplate Fluorometer) at an excitation wavelength of 480 nm and an emission wavelength of 530 nm. The exact FITC-dextran concentration in the blood was calculated using a standard curve with serially diluted FITC-dextran solutions. 

### 2.7. Assessment of Intestinal MPO Activity and Cytokine Levels in Mice

The activity of MPO, a heme-containing peroxidase expressed mainly in neutrophils, was measured in murine colonic tissue as a surrogate for neutrophil infiltration as described before [7]. Briefly, murine colonic samples were homogenized in potassium phosphate (pH 6.0) containing 0.5% hexadecyltrimethylammonium bromide (0.02 mL/mg tissue). Thereafter, the samples were homogenized, subjected to two freeze–thawing cycles and subsequently centrifuged at 15,000 rpm for 15 min at 4 °C. An aliquot (0.1 mL) of the supernatant was then added to 2.9 mL of o-dianisidine solution (i.e., 16.7 mg of o-dianosidine dihydrochloride in 1 mL of methyl alcohol, 98 mL of 50 mM potassium phosphate buffer at pH 6.0 and 1 mL of 0.005% H_2_O_2_ solution). Immediately afterwards, the change in absorbance of the samples was read at 460 nm over 60 s using a Spectronic Genesys 5 spectrophotometer (Milton Roy, Ivyland, PA, USA). One unit of MPO activity equals the amount of enzyme able to convert 1 mmol of H_2_O_2_ to H_2_O per min at 25 °C.

ELISA was performed to quantify the murine colonic protein levels of IL-1β, TNF, IL-6 and IL-22. To do so, snap-frozen colonic tissues were homogenized in ice cold NP-40 buffer (i.e., 20 mM Tris HCl [pH 8.0], 137 mM NaCl, 10% glycerol, 1% nonidet-40, 2 mM EDTA) supplemented with protease and phosphatase inhibitor cocktail tablets (Roche, Basel, Switzerland). After centrifugation (14.000 rpm, 10 min, 4 °C) to remove cell debris, the protein concentration was determined using the Pierce BCA protein assay kit (Thermo Fisher Scientific, Waltham, MA, USA). Thereafter, the cytokine levels were determined by analyzing 100 µL of a 2.5 µg/mL protein solution of each sample in duplicate using the mouse uncoated ELISA kits (88-7013-22 [IL-1β], 88-7324-22 [TNF], 88-7064-22 [IL-6], 88-7422-22 [IL-22], Invitrogen, Thermo Fisher Scientific, Waltham, MA, USA) according to the manufacturer’s instructions. On each plate a standard curve was run and absorbance at 450 nm was evaluated to allow protein quantification.

### 2.8. Patients and Clinical Specimens

Patients that underwent an endoscopy for clinical reasons (i.e., a suspicion or follow-up of IBD or a positive immunochemical Fecal Occult Blood Test [iFOBT]), were recruited via the policlinic of the University Hospital of Antwerp (UZA), a tertiary hospital in Belgium from 2018 to 2022. During endoscopy, biopsies from macroscopically inflamed and non-inflamed colonic regions were collected and stored in RNA later at −80 °C or embedded in paraffin until further use. A total of 14 IBD patients (7 UC and 7 CD) and 4 controls (with no macroscopic or microscopic abnormalities) were selected based on the presence (or absence in case of controls) of active colonic disease to investigate the expression of *MUC13* mRNA and important intestinal barrier-related and regulatory genes. Regarding CD patients, four patients showed active inflammation in both the ileum and colon, with two having disease predominantly in the ileum. Most patients (11/14) received active treatment for their disease. All subjects gave their informed consent for inclusion before they participated in the study. The study was conducted in accordance with the Declaration of Helsinki, and the protocol was approved by the Ethics Committee of UZA, Belgium (Belgian Registration number B300201733423 and B3002020000162).

### 2.9. RNA Extraction and RT-qPCR for Gene Expression

Total RNA was extracted from cells, murine tissue or human biopsies using the NucleoSpin^®^ RNA plus kit (Macherey-Nagel, Dueren, Germany) following the manufacturer’s instructions. The concentration and quality of the RNA were evaluated using the NanoDrop^®^ ND-1000 UV-Vis Spectrophotometer (Thermo Fisher Scientific, Waltham, MA, USA). One µg RNA was converted to cDNA by reverse transcription using the SensiFast™ cDNA synthesis kit (Bioline, Cincinnati, OH, USA). Relative gene expression was then determined by SYBR Green RT-qPCR using the GoTaq qPCR master mix (Promega, Leiden, The Netherlands) on a QuantStudio 3 Real-Time PCR instrument (Thermo Fisher Scientific, Waltham, MA, USA). Primer sequences are shown in Appendix A. All RT-qPCR reactions were performed in duplicate and involved an initial DNA polymerase activation step for 2 min at 95 °C, followed by 40 cycles of denaturation at 95 °C for 15 s and annealing/extension for 1 min at 60 °C. Analysis and quality control were performed using qbase+ software (Biogazelle, v3.4). Relative expression of the target genes for each sample was normalized to the geometric mean of the expression of the housekeeping genes *Actb* and *Rpl4* (for the murine samples) and *ACTB* and *GAPDH* (for the human samples) for that sample.

### 2.10. Illumina Short-Read Transcriptome Sequencing and Data Analysis

Prior to library preparation, the RNA concentration was measured on a Qubit fluorometer using a Qubit™ RNA Broad Range (BR) Assay Kit (Q10211, ThermoFisher, Waltham, MA, USA), and RNA purity and integrity was determined using a Nanodrop ND-1000 UV-Vis Spectrophotometer and 5300 Fragment Analyzer System (Agilent Technologies, Santa Clara, CA, USA). RNA samples extracted from *MUC1*, *MUC13* or ctrl siRNA transfected LS513 cells treated with IL-22 or vehicle (4 replicates for each condition of at least 3 independent experiments), were prepared with the QuantSeq 3′ mRNA-Seq Library Prep Kit FWD for Illumina (Lexogen GmbH, Vienna, Austria) following the standard protocol for long fragments. The resulting indexed cDNA libraries were equimolarly pooled and loaded for one NextSeq 500 sequencing run (high output v2 kit, 150 cycles, single read, Illumina, San Diego, CA, USA).

RNA samples extracted from the human biopsies (UC [N = 7], CD [N = 7] and control [N = 4]) were used to generate indexed cDNA libraries (NEBNext Ultra II Directional RNA library prep kit for Illumina, New England BioLabs, Ipswich, UK) which were then subsequently equimolarly pooled and loaded on a NovaSeq 6000 for sequencing (NovaSeq 6000 Reagent kits, 150 cycles, paired end, Illumina).

To analyze the obtained raw sequencing data sets, FastQC v0.11.9 and Trimmomatic v0.38 were used for quality control and read trimming, respectively [23]. The reads were mapped to the human reference genome build 38 using STAR v2.6.1a and read count tables were generated using featureCounts v2.0.0. [24,25]. Eventually, DEGs were identified using the Bioconductor package DESeq2 (v1.30.1) [26].

### 2.11. Western Blotting and ELISA for the Quantification of Human Intestinal Barrier Mediators

Forty micrograms of protein, extracted from LS513 cells using ice cold NP-40 buffer, was loaded on NuPAGE 12% Bis-Tris Protein Gels (Invitrogen, Thermo Fisher Scientific, Waltham, MA, USA). After gel electrophoresis, total protein was transferred onto an Immobilon-FL PVDF membrane (Sigma-Aldrich, Merck, Darmstadt, Germany). The blots were incubated overnight at 4 °C with the following primary antibodies: CLDN1 (1:200; Abcam, Cambridge, UK), CDH1 (1:1000; Cell Signaling Technology, Danvers, MA, USA), ROCK2 (1:2000; Invitrogen) and β-actin (1:20,000; Cell Signaling Technology). Target protein expression was visualized using infrared target detection by incubating the blots for 1 h at room temperature with IRDye 800CW Goat anti-Mouse (926-32210, Li-Cor, Lincoln, OR, USA) and IRDye 680RD Goat anti-Rabbit (926-68071, Li-Cor) secondary antibodies. The Odyssey Imaging system (Li-Cor), the Li-Cor Image studio v4.0 software and ChemiDoc Touch imaging software (BioRad, Hercules, CA, USA) were used for blot visualization and protein quantification. Β-actin was used for normalization of relative protein expression. ELISA was performed to quantify the human colonic protein levels of JAK1 (Abcam), phospho-JAK1 (pJAK1; Abcam), phospho-STAT3 (pSTAT3; ThermoFisher), phospho-MAPK1 (pMAPK1; ThermoFisher), phospho-MAPK9 (pMAPK9; ThermoFisher), SNAI1 (BioConnect, Huissen, The Netherlands) and ZEB1 (BioConnect). Protein levels were determined according to the manufacturer’s instructions provided with the ELISA kits. A standard curve was included and absorbance at 450 nm evaluated to allow protein quantification.

### 2.12. Immunohistochemistry of Human and Murine Colonic Samples

Five µm cross-sections were deparaffinized, rehydrated and used for immunohistochemical staining using target specific primary antibodies and visualization with a secondary streptavidin-horseradish peroxidase antibody and 3-amino-9-ethylcarbazole (AEC) substrate to detect the expression and localization of MUC1 (human: AF6298, 1:500, R&D systems, Minneapolis, MN, USA; mouse: ab15481, Abcam, 1:200) and MUC13 (human: MABC209, Sigma-Aldrich, Merck, 1:1000; mouse: in-house, 1:2000). The stained sections were analyzed by light microscopy (Olympus BX43).

### 2.13. Statistics

Statistical analysis using the GraphPad Prism 8.00 software (license DFG170003) was performed to determine significant differences between different experimental groups using Student’s T- and Analysis of Variance (ANOVA) tests. Non-parametric tests (Mann-Whitney U and Kruskal-Wallis tests) were used when appropriate. Concerning the RNA sequencing data analysis, R studio (R version 4.0.5) was used to perform a DESeq2 analysis that fits negative binomial generalized linear models for each gene and uses the Wald test for significance testing. Data are presented as means ± standard error of the mean (SEM), as min to max values in boxplots or as mean log 2-fold expression values in heatmaps, unless stated otherwise. *p*-values below 0.05 were considered statistically significant after correcting these values for multiple testing by using Tukey’s or Dunn’s multiple comparisons test or the Benjamini and Hochberg method (RNAseq data). All authors had access to the study data and had reviewed and approved the final manuscript.

## 3. Results

### 3.1. Silencing of MUC1 and MUC13 mRNA Expression Alters the Gene Expression of Several Intestinal Epithelial Barrier Mediators in the Presence/Absence of Inflammation

To unravel the mechanism(s) involved in *MUC13*-mediated barrier dysfunction and the additive effect of *MUC1* signaling upon inflammation, we first investigated the transcriptomes of LS513 IECs transfected with *MUC1*, *MUC13* or negative control siRNAs and subsequently treated with IL-22 or vehicle. Transfection resulted in a mean knockdown efficiency of 70% for *MUC13* and 75% for *MUC1* in untreated cells, as assessed by qPCR prior to sequencing (Figure 1A). Silencing of *MUC1* and *MUC13* mRNA strongly affected gene expression in untreated IECs and this effect was even more pronounced in IL-22 treated cells (Figure 1B). In addition, the effect of *MUC13* knockdown was the most pronounced in untreated cells (2730 vs. 1261 differentially expressed genes [DEGs]), whereas *MUC1* knockdown showed a larger effect in IL-22 treated cells (7933 vs. 5562 DEGs) (Figure 1B). Furthermore, mucin- and cytokine-specific effects on gene expression were also found, yet significant overlaps existed (Figure 1C). Subsequently, we evaluated the expression of 175 genes involved in intestinal barrier function and classified as genes encoding mucins, tight junctions, adherens junctions, (hemi)desmosomes, cytoskeleton proteins, cell polarity proteins, regulating proteins and proteins involved in epithelial cell immunity. These genes were selected based on the list of Vancamelbeke et al. [6] and further adapted based on an extensive literature search and their expression in LS513 cells (Appendix A). Silencing of *MUC13* and *MUC1* mRNA expression affected the expression of genes derived from all the abovementioned classes in the presence and absence of inflammation, as shown in Figure 1D. More specifically, *MUC13* deficiency significantly affected the differential expression of 35 barrier-related genes in untreated cells and five in IL-22 treated cells compared to control siRNA transfected cells (Figure 1D). Validation of a subset of these genes by qPCR revealed six additional DEGs in untreated and seven in IL-22 treated cells (Figure 1D). With regards to *MUC1* siRNA-transfected cells compared to ctrl siRNA transfected cells, significant changes in the expression of 13 barrier-related genes were noticed in untreated cells with two additional DEGs identified upon qPCR validation, whereas 50 genes were significantly altered in expression upon IL-22 stimulation with one additional significantly altered gene identified by qPCR (Figure 1D). Interestingly, IL-22 activation associated with 55 DEGs in control siRNA transfected cells and several of these significant alterations were not maintained upon *MUC13* knockdown (Figure 1D). Of note, *MUC13* deficiency also significantly affected *MUC1* expression during inflammation (Figure 1D). All these results clearly highlight that both MUC1 and MUC13 are important players in modulating intestinal barrier function, both under normal physiological as well as inflammatory conditions, and that these two specific mucins have a cooperating role.

### 3.2. Aberrant Expression of MUC1 and MUC13 Is Mainly Regulated by the JAK1/STAT3 Pathway upon IL-22 Activation

To explore which upstream transcriptional regulators are involved in the upregulation of *MUC1* and *MUC13* transcription upon IL-22 stimulation, activation z-scores were calculated for all regulatory proteins based on the overall changes in gene expression upon IL-22 stimulation in ctrl siRNA transfected cells using the Ingenuity Upstream Regulator Analysis tool from the Ingenuity Pathway Analysis (IPA) software (Figure 2A). Based on these calculations, SMARCA4, CEBPA, CEBPB, NF-κB, JAK/STAT and p38 MAPK were selected for further investigation using selective inhibitors or specific siRNAs (Figure 2B).

Blocking JAK activity by baricitinib (JAK1/JAK2 inhibitor) treatment significantly decreased the IL-22-induced *MUC1* and *MUC13* mRNA expression levels (Figure 2B). Inhibition of STAT3 using stattic (which inhibits activation, dimerization and nuclear translocation of STAT3) diminished *MUC1* mRNA expression (*p* = 0.08), yet enhanced *MUC13* mRNA expression (Figure 2B). On the contrary, suppression of STAT1 via fludarabine treatment (inhibits STAT1 phosphorylation) did not affect *MUC1* nor *MUC13* mRNA expression (Figure 2B). Interestingly, simultaneous *MUC1* and *MUC13* knockdown decreased the phosphorylation of both JAK1 and STAT3, indicating a mutual regulatory effect between these mucins (Figure 2C–F).

In addition, downregulation of the p38 MAPK signaling pathway using N-feruloyloctopamine also resulted in a significant decrease of *MUC13* mRNA expression during IL-22 stimulation (*p* < 0.05; Figure 2B), whereas a significant increase in *MUC1* mRNA expression was noted when IL-22-stimulated cells were transfected with *SMARCA4* or *CEBPB* siRNA (Figure 2B). Interfering with NF-κB activity using BAY 11-7085 did not affect *MUC1* nor *MUC13* expression. The findings above clearly indicate a complex regulation of *MUC1* and *MUC13* transcription in IECs via several signaling pathways activated by IL-22.

### 3.3. Aberrant MUC13 Expression Might Affect Intercellular Junctions in IECs via SNAI1/ZEB1 and ROCK2/MAPK Signaling Pathways upon IL-22 Activation

To investigate which downstream regulators are involved in MUC13-mediated intestinal barrier dysfunction, we first determined transepithelial electrical resistance (TEER) and fluxes of 4 kDa fluorescein isothiocyanate (FITC)-dextran in control and *MUC13* siRNA transfected LS513 cells upon IL-22 stimulation (Figure 3A,B). Paracellular permeability to 4 kDa FITC-dextran was significantly increased in control siRNA transfected cells upon IL-22 treatment, whereas TEER was decreased (Figure 3A,B). Contrary to this, silencing of *MUC13* mRNA expression significantly decreased the 4 kDa FITC-dextran permeability in treated and untreated cells, and increased the TEER in untreated cells (Figure 3A,B). Furthermore, epithelial barrier integrity was also studied at the molecular level in LS513 cells transfected with *MUC13* siRNA or *MUC1*/*MUC13* siRNA followed by IL-22 treatment to verify the transcriptomic dataset (Figure 1D) and to investigate the additive role of *MUC1* signaling (Figure 3). *MUC13* knockdown significantly affected mRNA expression of *CLDN1*, *CLDN3*, *CLDN4*, *TJP2* (i.e., *ZO-2*), *PRKCZ, CRB3, SCRIB, CEACAM1, CEACAM6* and *ACE2* in the absence or presence of inflammation (Figure 3E,F,I,K,L,N,P–S,U,V). The simultaneous knockdown of both *MUC1* and *MUC13* maintained the significant alterations of *CLDN1*, *CEACAM1* and *ACE2* (Figure 3E,F,S,V) and induced an effect on *CDH1* and *OCLN* expression upon IL-22 stimulation (Figure 3G,O). Interestingly, simultaneous knockdown of both *MUC1* and *MUC13* affected the expression of *IL10RB* (i.e., a subunit of the IL22 receptor [IL22R]), suggesting the presence of a feedback loop (Figure 3W). 

Subsequently, JAK1, MAPK1, MAPK9, ROCK2, SNAI1 and ZEB1 were, based on their changing mRNA expression pattern due to *MUC13* knockdown (Figure 1D) and for their role in epithelial barrier homeostasis [4,27,28,29], further studied as potential downstream regulators of *MUC13*-mediated barrier dysfunction (Figure 2C,D and Figure 4). *MUC13* deficiency induced an elevated expression of *MAPK1*, *ROCK2*, *SNAI1* and *ZEB1* mRNA upon IL-22 stimulation, whereas simultaneous *MUC1* and *MUC13* knockdown abolished these effects (Figure 4A,E,G,I). Aberrant phosphorylation patterns of JAK1, MAPK1 and MAPK9, and altered protein expression patterns of ROCK2 upon *MUC13* knockdown and/or simultaneous knockdown of *MUC1* and *MUC13* were also observed (Figure 2E and Figure 4B,D,F,H). Although these qPCR results failed to recapitulate the effect of *MUC13* knockdown initially observed in our transcriptome dataset, these results further indicate a potential role for MUC13 in the modulation of the expression/activity of these key barrier regulators, which also seems to be partially mediated through *MUC1* signaling.

### 3.4. Muc13^−/−^ Mice Show a Differential Response to DSS-Incuded Colitis

We next verified the above obtained in vitro IEC data using *Muc13* deficient mice generated via the CRISPR-Cas9 technology. The *Muc13* knockout efficiency was first validated by RT-qPCR and IHC (Figure 5). No differences in weight or overall development were observed between wildtype and *Muc13*^−/−^ mice (Appendix A).

Mice were then challenged with repeated cycles of 2–3% DSS in their drinking water to induce acute (i.e., first cycle of DSS administration) and chronic (i.e., second and third cycle of DSS administration) colitis, during which inflammation and barrier functionality were studied. *Muc13*^−/−^ mice seemed to exhibit more severe clinical signs of colitis and macroscopic colonic inflammation than wildtype controls upon acute DSS colitis, whereas an enhanced recovery was noticed after the second cycle of DSS administration (Figure 6A–D). Nevertheless, no statistical differences between *Muc13*^−/−^ and wildtype mice were identified. Furthermore, the colon weight/length ratio, the spleen weight, myeloperoxidase (MPO) activity and protein expression of tumor necrosis factor (TNF), IL-1 β, IL-6 and IL-22 were significantly higher in the DSS-treated colons of both wildtype and *Muc13*^−/−^ mice during acute and chronic colitis (Figure 6E–H), with only a significant increase in TNF protein levels due to *Muc13* knockdown after the first cycle of DSS administration compared to the control (Figure 6H). 

### 3.5. Aberrant Muc13 Expression Affects Intestinal Barrier Function in DSS-Treated Mice

Results of the FITC-dextran intestinal permeability assay showed that the integrity of the mucosal barrier was already affected in the colon of control *Muc13*^−/−^ mice (Figure 7A), although no inflammatory abnormalities were observed (Figure 6). Furthermore, FITC-dextran intestinal permeability was strongly increased after the first cycle of DSS administration in wildtype mice, after which it declined in the chronic stages of colitis with only a significant increase after the second DSS cycle (Figure 7A). On the contrary, DSS-treated *Muc13*^−/−^ mice showed only a significant increase in intestinal permeability after the first cycle compared to their control counterparts, but no significant alterations in barrier function were noted between DSS-treated wildtype and *Muc13*^−/−^ mice (Figure 7A). *Muc1* and *Muc13* signaling was significantly increased in wildtype mice upon DSS-administration (Figure 7B,C), with MUC13 showing positive staining in the cytoplasm in particular after the third cycle of DSS administration (Figure 7C). In addition, expression of *Muc1* was measured in *Muc13*^−/−^ mice to investigate whether there was a compensatory effect of this mucin. A significant increase of *Muc1* mRNA expression due to *Muc13* knockdown was only seen after the first DSS cycle (Figure 7B), whereas a more pronounced protein expression was observed in the cytoplasm of IECs upon the second DSS cycle (Figure 7C). Subsequently, intestinal barrier function was also investigated at the molecular level. mRNA expression of *Cldn1* and *Cldn2* was significantly altered in control *Muc13*^−/−^ mice compared to wildtypes and during the course of colitis (Figure 7B,C and Appendix A). *Jam2* and *Ocln* mRNA expression was significantly increased in both wildtype and *Muc13*^−/−^ mice upon acute and chronic colitis, respectively, whereas *Tjp2* mRNA expression was only elevated in *Muc13*^−/−^ mice after the first DSS cycle (Figure 7B and Appendix A). Expression of *Cldn3*, *Cldn4*, *Cldn7* and *Cdh1* mRNA was however not altered during colitis or due to the absence of *Muc13* expression in mice (Figure 7B and Appendix A). Differences in mRNA expression of cell polarity subunits were only observed for *Scrib* and aPkcz after the third cycle of DSS administration in both wildtype and *Muc13*^−/−^ mice whereas expression of *Ceacam 1*, and *Tlr4* mRNA was found to be significantly altered upon chronic colitis (Figure 7B and Appendix A). Finally, the expression of the upstream and downstream regulators of Muc13 signaling, as identified above, were also evaluated. In control animals, *Mapk9* mRNA was upregulated in the colon due to the absence of *Muc13* expression, whereas the mRNA levels of *Mapk1*, *Mapk9*, *Rock2* and *Snai1* were also affected upon acute colitis in *Muc13*^−/−^ mice compared to their wildtype counterparts (Figure 7B and Appendix A). Expression of *Zeb1* and *Jak1* mRNA was also significantly increased upon acute DSS colitis, although no difference in expression was noted between DSS-treated *Muc13*^−/−^ and wildtype mice (Figure 7B and Appendix A). In addition, mRNA expression of *Il10rb* and *Il22ra* was significantly upregulated upon the second DSS cycle (Figure 7B and Appendix A).

### 3.6. Expression of Intestinal MUC13 and Its Barrier Mediators Alters upon Inflammation in IBD

We further validated these in vitro and in vivo results translationally using biopsies from IBD and control patients. Colonic mucosal biopsies from 26 IBD patients (15 UC and 11 CD) and ten healthy individuals were first independently blinded and scored by a pathologist assessing inflammation and subsequently analyzed for MUC13 expression. Expression of *MUC13* mRNA was significantly upregulated in the colonic mucosa from IBD patients compared to controls (Figure 8A). At protein level, MUC13 appeared to be predominantly limited to the glycocalyx at the luminal surface in the mucosa of healthy individuals, which became more pronounced in IBD patients in addition to its expression in the cytoplasm of colonic enterocytes (Figure 8B). Additionally, MUC1 expression was significantly increased both at RNA and protein level in the colon of IBD patients compared to controls (Figure 8A,B) and the expression patterns of both mucins did not seem to differ between CD and UC patients (Figure 8A,B). Finally, we assessed colonic mRNA expression of barrier proteins, IL-22 and its receptor involved in MUC13-mediated barrier dysfunction (Figure 8C,D). mRNA Expression of *IL-22*, *IL10RB*, *CLDN1*, *CLDN2*, *CLDN7*, *CDH1*, *JAK1*, the *SNAI1/ZEB1* and *ROCK2/MAPK* signaling pathways and several *CEACAMs* was significantly altered in the inflamed colon of IBD patients compared to the healthy mucosa of controls (Figure 8C). Distinct gene expression patterns were observed among the UC, CD and control groups, but further experiments are needed to investigate associations between the expression of MUC13 and other genes involved in barrier function in IBD patients (Figure 8C,D).

## 4. Discussion

This is the first study providing evidence on how aberrant MUC13 expression affects intestinal epithelial barrier function upon IL-22 activation (summarized in Figure 9). In our study, we showed that the absence of MUC13 altered the gene expression of several tight junctions as main regulators of paracellular permeability [27,28,29,30,31]. Particularly, a significant upregulation of *CLDN1*, *CLDN3* and *CLDN4* was noted in *MUC13* siRNA transfected IECs whereas a significant alteration in mRNA expression was also shown for *CLDN1*, *CLDN4, CLDN7* and *TJP2* upon IL-22 activation. Although similar changes in tight junction expression were seen upon *MUC1* knockdown, deficiency of both mucins in IECs mainly impacted on CLDN1 and CDH1 expression highlighting a major functional link between MUC1, MUC13, CLDN1 and CDH1 expression (Figure 9). The latter two proteins have major sealing characteristics in the intestinal epithelial barrier and their aberrant expression has been reported in IBD [30,31,32,33]. In addition to junctional proteins, knocking down *MUC13* expression with or without simultaneous *MUC1* silencing also affected the expression of several polarity proteins (*CRB3*, *SCRIB*, *PRKCZ*) and the epithelial cell receptor *ACE2* in untreated IECs, as well as other cell receptors regulating cell homeostasis, such as CEACAM1, CEACAM6 and TLR4, upon IL-22 stimulation (Figure 9). Expression of ACE2, CEACAMs and TLR4 is altered in the intestinal mucosa of IBD patients [34,35]. Loss of ACE2 has been associated with intestinal barrier dysfunction and aggravated colitis severity [36,37], whereas a strong correlation between *ACE2* and *MUC13* has been shown in airway epithelial cells and intestinal metaplastic cells of the stomach [38,39]. Similarly, *Muc1* knockdown also affected *Ace2* expression during gastric infection [40]. Furthermore, CEACAMs and TLR4 play an important role in regulating epithelial immunity by serving as bacterial adhesion molecules and have also been shown to affect intestinal barrier integrity upon inflammation [34,35,37,41,42,43,44]. In addition, aberrant *MUC1* and *MUC13* expression modulate IL-8 secretion in gastrointestinal epithelial cells, a downstream effect of activated TLR4 expression [11], which further indicates an interconnection of their signaling pathways [11]. The above findings clearly highlight the mutual regulation between transmembrane mucins and other cell receptors like ACE2, CEACAMS and TLRs (Figure 9).

Dysregulation of the abovementioned barrier mediators due to *MUC13* deficiency was also associated with an overall decreased paracellular permeability to 4 kDa FITC-dextran and increased TEER in IECs in the presence/absence of IL-22 activation. These results position MUC13 as an important participant in the dysfunction of the intestinal mucosal barrier [7]. Some changes in junctional protein expression were also noted in the absence of Muc13 before DSS exposure in vivo. More specifically, an increase in the expression of *Cldn1* and *Cldn2* mRNA was found in *Muc13*^−/−^ control mice. However, the impact of *Muc13* deficiency on tight junction expression was abolished upon DSS administration. A similar phenomenon was also seen when measuring intestinal permeability in vivo. Whereas the integrity of the mucosal barrier was affected due to *Muc13* deficiency in the colon of control mice, no significant alterations in barrier function were noted between DSS-treated wildtype and *Muc13*^−/−^ mice. Although *Muc1* was significantly increased during the course of colitis in DSS-treated mice, a compensatory increase in *Muc1* expression was also seen upon acute colitis in *Muc13*^−/−^ mice, which could explain the abolishment of the *Muc13* knockdown effect on the barrier function upon DSS exposure. Furthermore, the absence of *MUC13* also affected the expression of *MUC1* upon IL-22 activation in IECs and we previously described strong positive correlations between *Muc1* and *Muc13* expression in DSS colitis mice [7] and an IBD cohort [21]. Our data clearly suggest that MUC1 and MUC13 have cooperating regulated barrier roles, which contrasts with previous findings [8,11]. In addition, acute DSS treatment seemed to induce more severe clinical symptoms of colitis and higher macroscopic inflammation in the absence of MUC13, which is similar as described before [8]. However, in our study, we did not observe significant differences in the secretion of several inflammatory cytokines (TNF, IL-1β, IL-6 and IL-22) between wildtype and *Muc13*^−/−^ DSS-treated mice. Of note, the differences in colitis severity were primarily apparent during the recovery phase (i.e., no DSS treatment) whereas molecular assessment of the colon was made at the end of each DSS cycle, which could potentially explain the lack of significant differences in these inflammatory markers. On the contrary, our data suggested an enhanced recovery upon chronic-induced colitis in *Muc13*^−/−^ mice and a previous study even described that *MUC13* knockdown was associated with less tumor formation in AOM/DSS-treated mice [15]. These results also highlight the involvement of MUC13 in intestinal epithelial response to injury and inflammation.

As no direct interactions between mucins and intercellular junctions have been described so far, it is most likely that key regulatory proteins are involved in MUC13-induced mucosal barrier dysfunction upon IL-22 activation. Our data show that IL-22 triggers both MUC1 and MUC13 overexpression via activation of the JAK1/STAT3 pathway (Figure 9). IL-22 has multifaced roles in regulating cell proliferation, cell survival, wound healing, host defense and inflammation. This monomer cytokine does thus not only play an essential role in mucosal healing but also has deleterious effects on intestinal inflammation when uncontrolled [20]. The primary targets of IL-22 are IECs and stimulation of this cytokine induces the formation of the heterodimer complex of the IL-22 receptor comprised of IL-22R1 and IL-10R2 [45,46]. This results on its turn in the activation of JAK1 and TYK2, followed by the phosphorylation of STAT3 [20]. Induction of the JAK1/STAT3 pathway due to constitutive interaction with the IL-22R1 receptor in the intestinal epithelium plays a major role in IBD pathophysiology [47]. Our results showed that the simultaneous knockdown of MUC1 and MUC13 in IECs resulted in an increased IL10RB (IL-10R2) expression suggesting the existence of a certain feedback mechanism. Unlike MUC1, IL-22 also regulates MUC13 expression via p38 MAPK activation (Figure 9), a signaling pathway involved in TLR4-mediated cell proliferation and immune dysfunction in IBD [48]. Furthermore, a subset of IBD patients have elevated IL-22 expression levels (as also shown in our IBD cohort), which were reduced following anti-IL23 treatment resulting in high rates of clinical responses and remission, further suggesting that IL-22 is rather detrimental than protective in IBD [19]. The main downstream pathways found in our study and accounting for the mechanisms on how aberrant MUC13 expression affected junctional protein expression in IECs, are the SNAI1/ZEB1 and ROCK2/MAPK signaling axes (Figure 9). SNAI1 and ZEB1 are two major regulators of EMT, a phenomenon characterized by loss of apicobasal polarity and cell–cell contacts in epithelial cells [49]. SNAI1 has been demonstrated to induce ZEB1 expression and is a potential repressor of E-cadherin and several tight junctions (CLDN1, OCLN, ZO-1; Figure 9) in IECs thereby promoting intestinal barrier dysfunction upon colitis and tumorigenesis in mice [28,49,50,51]. Also in our study, we identified a link between MUC13 deficiency in combination with simultaneous *MUC1* silencing and a reduced E-cadherin expression, which is considered a key marker of EMT and a validated downstream target of ZEB1 [52], as well as significant alterations of *CLDN1* and *CLDN7* expression which have also been shown to promote EMT in IECs [53,54,55].

Furthermore, expression of MAPK1 (ERK2) and MAPK9 (JNK2) were both affected upon MUC13 knockdown in the intestinal tract upon inflammation (Figure 9). It has previously been shown that suppressing both MAPK pathways contributed to the amelioration of chemically induced murine colitis and the restoration of the intestinal barrier by elevating tight junction expression (CLDN1, ZO-1, OCLN) [56,57]. Similarly, Rho kinase (ROCK) inhibition also reduced intestinal barrier dysfunction during experimental colitis [29,58]. Previous studies highlighted a complex regulatory role of ROCK signaling on intercellular junction formation. On the one hand, ROCK activation upon inflammation has been reported to repress tight (ZO-1) and adherens (E-cadherin, β-catenin) junction expression in IECs through myosin light chain kinase (MLCK) [59,60]. On the other hand, a positive modulation of intercellular junctions by ROCK was also observed since inhibition decreased expression of CLDN1, OCLN, ZO-1 and E-cadherin [61,62]. This is similar to our findings, as we noticed a simultaneous upregulation of *ROCK2*, *CLDN1* and *CDH1* due to *MUC13* knockdown in IL-22 stimulated cells.

Nevertheless, we need to acknowledge that our study has several limitations. Firstly, upon IL-22 stimulation of LS513 cells, we noticed the upregulation of several inflammatory molecules (i.e., *IL1B*, *IL10*, *IL32* and *TGFB1*) known to signal through the JAK/STAT pathway and therefore we cannot rule out that such inflammatory proteins could also have affected the phosphorylation status of STAT3 and JAK1. Secondly, the molecular analyses in mice were performed on whole colonic tissue sections, implying that we did not only evaluate intestinal epithelial cells and did not account for potential differences in cell composition caused by the infiltration of immune cells or loss of epithelial cells. Thirdly, our molecular analyses mainly focused on changes in the expression of major barrier mediators at the mRNA and/or protein level, but it needs to be noted that intercellular junctions are dynamic structures consisting of multiple proteins and involving repeated assembly and remodeling, which may occur despite no clear differences in expression. In addition, the proteins found in the intercellular junctions such as claudins also localize to numerous other sites in the cell to exert other functions in epithelial cells, and in this way changes in their expression might not be solely related to the intercellular junction. 

In summary, we identified an important role for the involvement of MUC13 in intestinal barrier dysfunction by modulating numerous barrier-related genes in the presence/absence of IL-22 stimulation, with a major cooperating role for MUC1, thereby impacting intestinal barrier integrity as evidenced in IECs and our in vivo *Muc13*^−/−^ mice. Moreover, opposing effects on disease activity during acute and chronic DSS-induced colitis were noticed: the presence of MUC13 was protective during the acute phase whereas it was harmful during chronic DSS administration. IL-22-induced JAK/STAT signaling was clearly implicated in the regulation of MUC1 and MUC13 expression, further suggesting a potential use of JAK inhibitors to interfere with aberrant mucin expression and subsequent barrier function in the inflamed mucosa of IBD patients, as has also been highlighted in other diseases including cancer and COVID-19 [63,64,65,66,67]. Of note, as the pathophysiology of IBD involves mutual interactions of numerous inflammatory pathways that can affect mucin expression and function [7], future studies are encouraged to further investigate these pathways in relation to the cumulative effects of transmembrane mucin signaling on intestinal barrier function in IBD.

## Figures and Tables

**Figure 1 cells-12-01224-f001:**
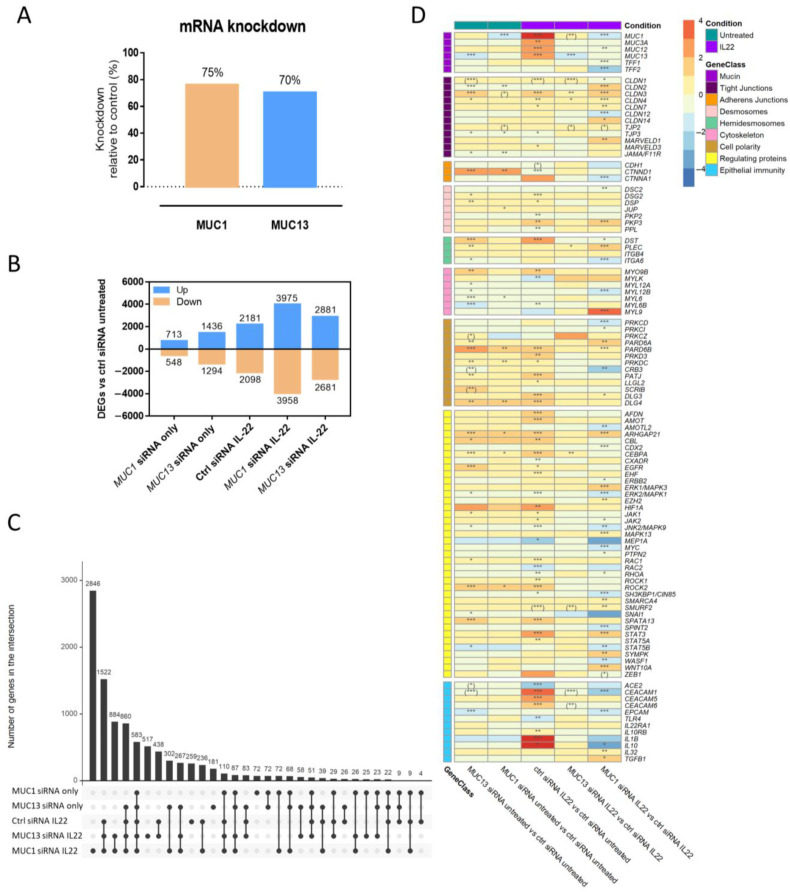
Landscape of differentially expressed genes involved in the modulation of intestinal barrier integrity in the presence/absence of *MUC1* or *MUC13* mRNA expression and IL-22 stimulation in intestinal epithelial cells. LS513 cells were transfected for 48 h with *MUC1*, *MUC13* or negative control (ctrl) siRNA, followed by IL-22 stimulation for 24 h upon reaching full confluence. (**A**) Percentage of knockdown in mRNA expression of *MUC1* and *MUC13* in LS513 cells (N ≥ 3 independent experiments) upon siRNA transfection for 72 h. (**B**) Number of significantly upregulated and downregulated genes for each condition relative to untreated ctrl siRNA-transfected cells (Benjamini Hochberg adjusted *p*-value < 0.05; N = 4/condition from at least three independent experiments). (**C**) Upset plot showing general features of differential gene expression analyses (DESeq2 analysis versus untreated ctrl cells) upon *MUC1* and *MUC13* knockdown during IL-22 stimulation. (**D**) Heatmap showing the log2 fold change expression data of genes that are involved in intestinal barrier homeostasis and that were significantly altered upon IL-22 stimulation and/or *MUC13* or *MUC1* knockdown in LS513 cells (* *p* < 0.05; ** *p* < 0.01; *** *p* < 0.001; N = 4/condition from at least three independent experiments). In addition, significantly altered genes that were identified by qPCR are indicated with brackets. The complete list of investigated genes can be found in Appendix A.

**Figure 2 cells-12-01224-f002:**
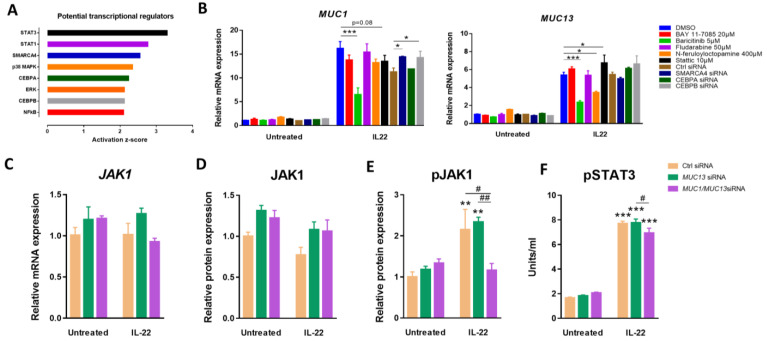
Upstream regulators of IL-22-induced *MUC1* and *MUC13* signaling. (**A**) Potential transcriptional regulators of *MUC1* and *MUC13* mRNA expression with activation z-scores > 2 as identified by overall changes in gene expression in IL-22-stimulated LS513 cells compared to untreated cells using the IPA Upstream Regulator Analysis. (**B**) Relative mRNA expression levels of *MUC13* and *MUC1* in LS513 cells treated with BAY11-7085 (NFkB), Baricitinib (JAK1), Fludarabine (STAT1), N-feruloyloctopamine (Akt/p38 MAPK), Stattic (STAT1) or transfected with target specific siRNAs (SMARCA4, CEBPA, CEBPB) prior to stimulation with 40 ng/mL IL-22 for 24 h. Significant differences between control and treated cells in the absence or presence of inflammation are indicated by * *p* < 0.05; ** *p* < 0.01; *** *p* < 0.001 (Two-Way ANOVA; N = 6/condition from three independent experiments). (**C**–**F**) Relative mRNA and protein expression of JAK1 (**C**,**D**) and relative protein expression of pJAK1 (**E**) and pSTAT3 (**F**) in LS513 cells treated with IL-22 for 24 h following transfection with *MUC13* or *MUC1*/*MUC13* siRNAs for 48 h. Untreated cells and cells transfected with a negative control siRNA were included as controls. Significant differences are indicated by * *p* < 0.05; ** *p* < 0.01; *** *p* < 0.001 versus untreated cells and by # *p* < 0.05; ## *p* < 0.01 versus ctrl siRNA (Two-Way ANOVA; N = 6/condition from three independent experiments).

**Figure 3 cells-12-01224-f003:**
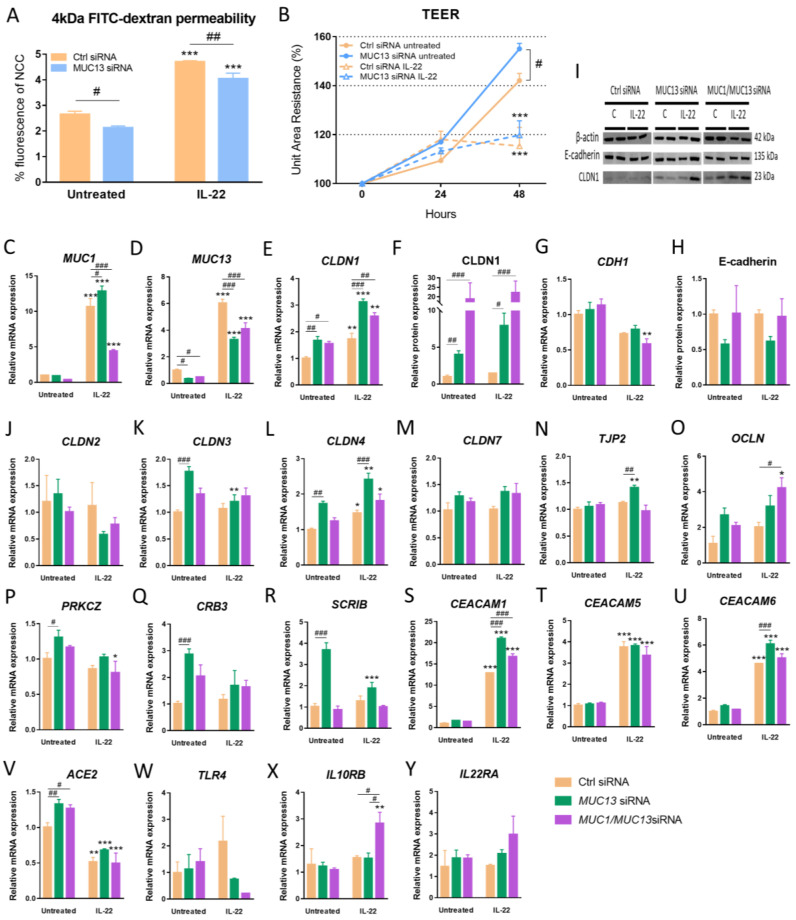
MUC13 impacts on intestinal barrier function in intestinal epithelial cells in presence/absence of IL-22 stimulation. (**A**) Analysis of 4 kDa FITC-dextran permeability and (**B**) evaluation of the transepithelial electrical resistance (TEER) in intestinal epithelial monolayers of LS513 cells that were transfected with *MUC13* siRNA for approximately 10 days (until TEER stabilized) followed by stimulation with 100 ng/mL IL-22 for 48 h. Untreated cells and cells transfected with a negative control siRNA were included as controls (N = 4 from two independent experiments). Relative mRNA expression of *MUC1* (**C**), *MUC13* (**D**), *CLDN1* (**E**), *CDH1* (E-cadherin); (**G**), *CLDN2* (**J**), *CLDN3* (**K**), *CLDN4* (**L**), *CLDN7* (**M**), *TJP2* (ZO-2); (**N**), *OCLN* (**O**), *PRKCZ* (**P**), *CRB3* (**Q**), *SCRIB* (**R**), *CEACAM1* (**S**), *CEACAM5* (**T**), *CEACAM6* (**U**), *ACE2* (**V**), *TLR4* (**W**), *IL10RB* (**X**) and *IL22RA* (**Y**) and relative protein expression levels, as defined by Western blotting, of CLDN1 (**F**,**I**) and E-cadherin (**H**,**I**) in LS513 cells treated with IL-22 for 24 h following transfection with *MUC13* or *MUC1*/*MUC13* siRNAs for 48 h. Untreated cells and cells transfected with a negative control siRNA were included as controls. Significant differences are indicated by * *p* < 0.05; ** *p* < 0.01; *** *p* < 0.001 versus untreated cells and by # *p* < 0.05; ## *p* < 0.01; ### *p* < 0.001 versus ctrl siRNA (Two-Way ANOVA, N = 6/condition from three independent experiments).

**Figure 4 cells-12-01224-f004:**
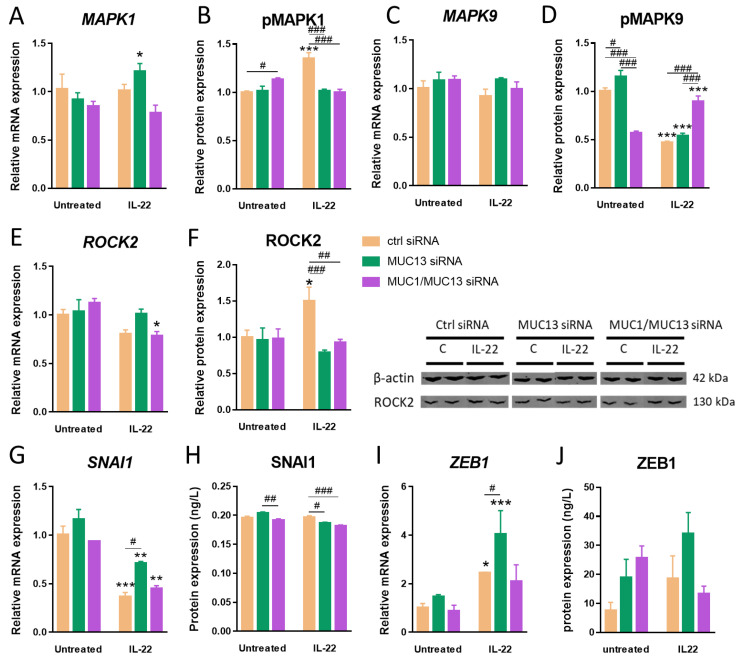
Downstream regulators of MUC13-mediated intestinal barrier dysfunction. Relative mRNA expression of *MAPK1* (**A**), *MAPK9* (**C**), *ROCK2* (**E**), *SNAI1* (**G**) and *ZEB1* (**I**) and relative protein expression of pMAPK1 (**B**), pMAPK9 (**D**), ROCK2 (**F**), SNAI1 (**H**) and ZEB1 (**J**) in LS513 cells transfected with *MUC13* (green) or *MUC1*/*MUC13* (purple) siRNAs 48 h prior to stimulation with 40 ng/mL IL-22 for 24 h. Untreated cells and cells treated with a negative control (ctrl; orange) siRNA were included as controls. Significant differences are indicated by * *p* < 0.05; ** *p* < 0.01; *** *p* < 0.001 versus untreated cells and by # *p* < 0.05; ## *p* < 0.01; ### *p* < 0.001 versus ctrl siRNA (Two-Way ANOVA, N = 6/condition from three independent experiments).

**Figure 5 cells-12-01224-f005:**
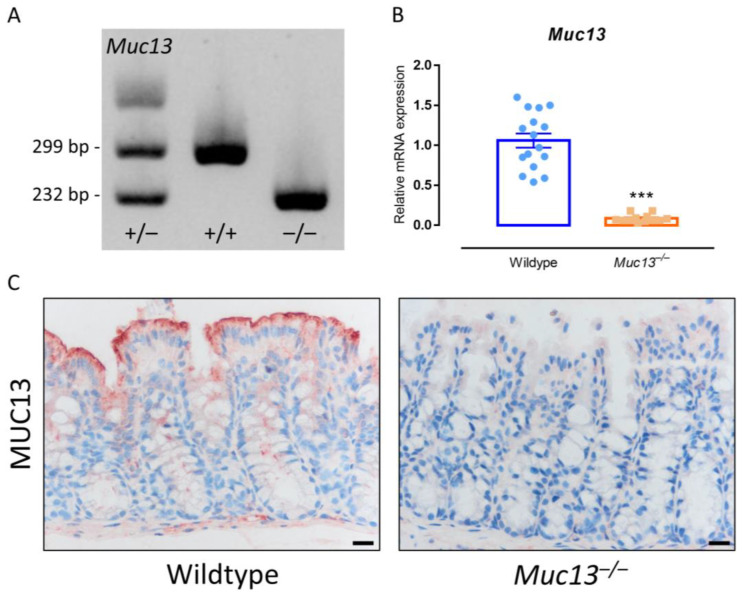
Validation of the *Muc13*^−/−^ mouse model. (**A**) Evaluation of CRISPR-Cas9 induced deletion in exon 4 of the *Muc13* gene by DNA gel electrophoresis. (**B**) Relative mRNA expression of Muc13 in the colon of healthy wildtype (WT, N = 16) and *Muc13*^−/−^ (N = 12) mice assessed by qPCR. Significant differences are indicated by *** *p* < 0.001 (Independent Student’s *t*-test). (**C**) Immunohistochemical analysis of MUC13 protein expression in the colon of healthy wildtype and *Muc13*^−/−^ mice. Representative images are shown. In *Muc13*^−/−^ mice, no positive staining for MUC13 can be found in the colonic epithelium.

**Figure 6 cells-12-01224-f006:**
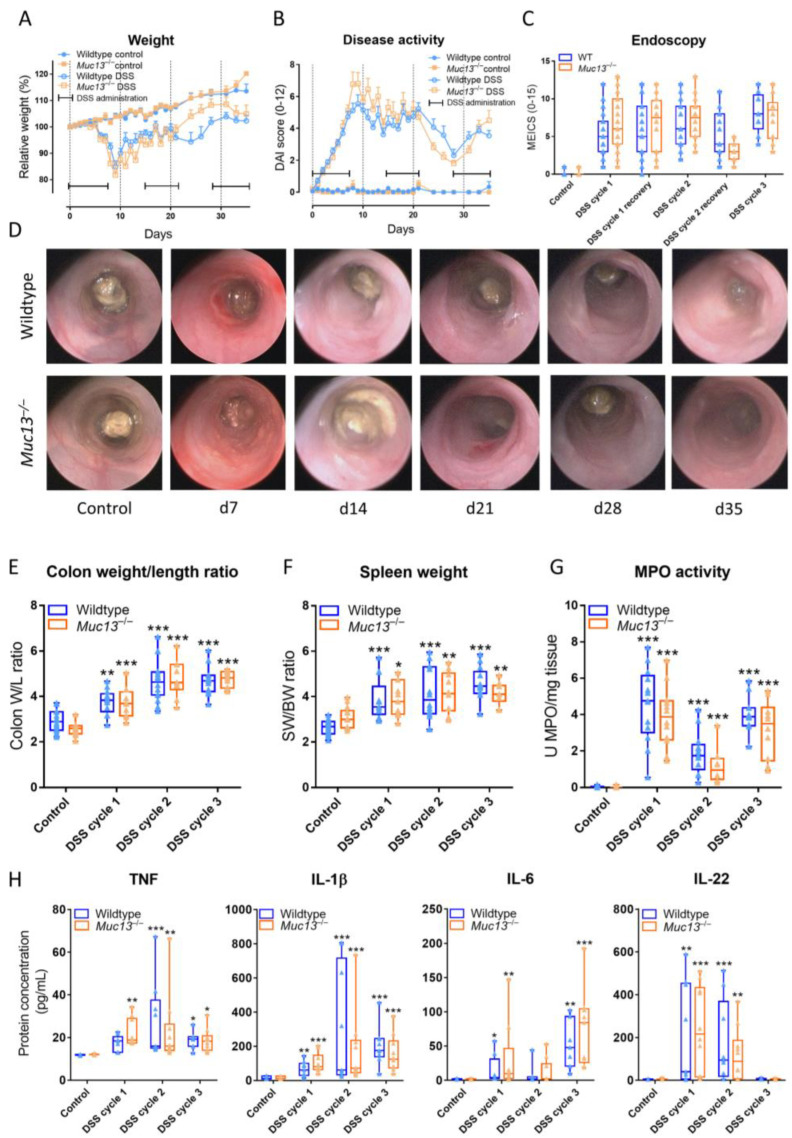
Disease activity and inflammatory parameters in wildtype and *Muc13*^−/−^ mice during the course of DSS-colitis. (**A**) Relative changes in weight. (**B**) Determination of the disease activity index (DAI), which is the cumulative score of body-weight loss, the extent of rectal bleeding and changes in stool consistency. Significant differences between wildtype and *Muc13*^−/−^ were tested by Kruskal-Wallis tests followed by Dunn’s post-hoc tests comparing the AUCs during the respective 7-day treatment and recovery cycles (N(controls) = 16–22/group, N(DSS) = 33–41/group). (**C**) Colitis severity was scored every week by endoscopy and was based on the cumulative assessment of thickening of the colon wall, changes in vascular pattern, presence of fibrin, mucosal granularity and stool consistency resulting in the murine endoscopic index of colitis severity (MEICS). Significant differences between wildtype and *Muc13*^−/−^ were tested using Mann-Whitney U tests (N = 43–50/group). (**D**) Representative images taken during colonoscopy. (**E**) The colon weight/length ratio as a macroscopic measure of colitis severity (N = 10–18/group). (**F**) Spleen weight (N = 10–19/group). (**G**) MPO activity (N = 10–21/group). (**H**) Colonic protein expression of TNF, IL-1β, IL-6 and IL-22 evaluated by ELISA (N = 8–10/group). Significant differences between control and colitis mice are indicated by * *p* < 0.05; ** *p* < 0.01; *** *p* < 0.001 (Two-Way ANOVA).

**Figure 7 cells-12-01224-f007:**
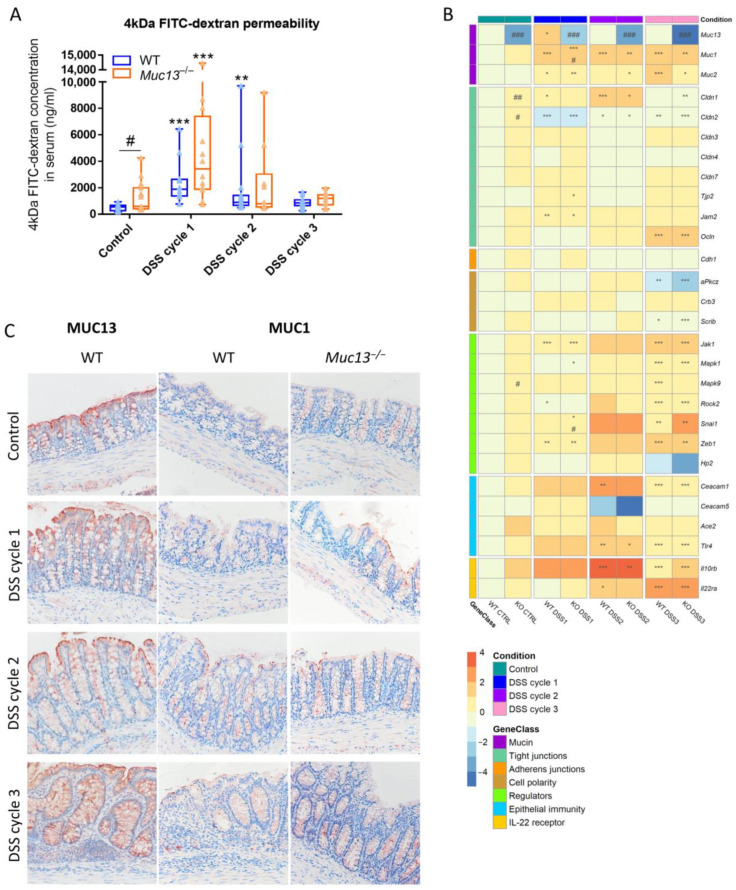
Analysis of epithelial barrier integrity in wildtype and *Muc13*^−/−^ mice during the course of DSS-colitis. (**A**) Analysis of 4 kDa FITC-dextran intestinal permeability in the DSS-induced colitis model (N = 10–20/group). (**B**) Heatmap visualization of the relative mRNA expression of mucins (*Muc1*, *Muc13*), tight junctions (*Cldn1, Cldn2, Cldn3, Cldn4, Cldn7, Jam2, Tjp2, Ocln*), adherens junctions (*Cdh1*), cell polarity subunits (*aPkcz, Crb3, Scrib*), regulators of intestinal barrier function (*Jak1*, *Mapk1*, *Mapk9, Rock2, Snai1, Zeb1, Hp2* (Zonulin)) and genes involved in epithelial immunity (*Ceacam1, Ceacam5, Ace2, Tlr4, Il10rb, Il22ra*) in the colon of wildtype (WT) and *Muc13*^−/−^ (KO) mice during the course of DSS-colitis (N = 7–17/group/gene). (**C**) Immunohistochemical analysis of MUC1 and MUC13 expression in the colon of healthy and DSS-treated WT and *Muc13*^−/−^ mice. Representative images were selected. Scale bars are 20 μm. Significant differences between control and colitis mice are indicated by * *p* < 0.05; ** *p* < 0.01; *** *p* < 0.001 and between WT and *Muc13*^−/−^ are indicated by # *p* < 0.05; ## *p* < 0.01; ### *p* < 0.001 (Two-Way ANOVA).

**Figure 8 cells-12-01224-f008:**
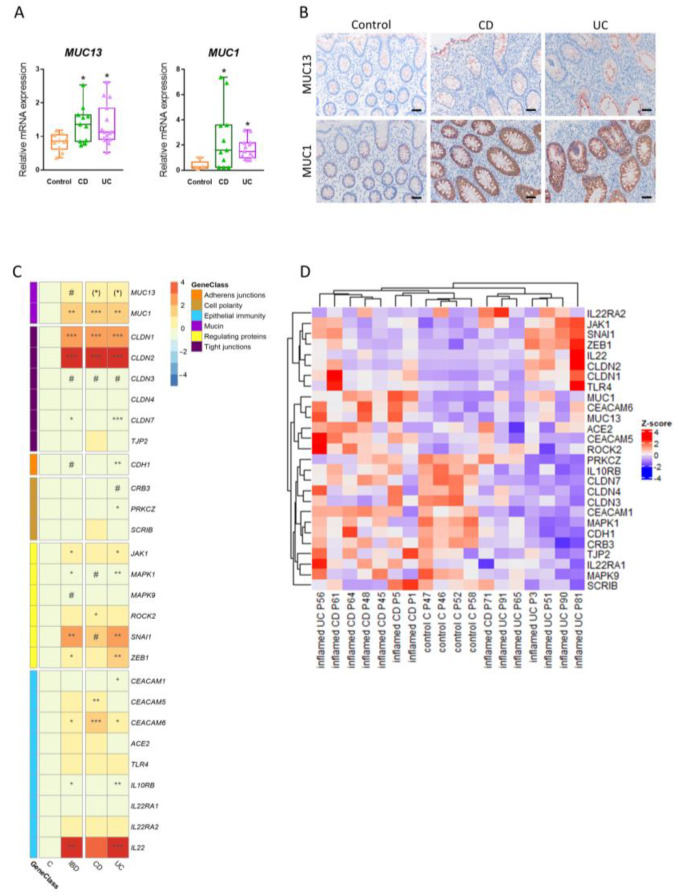
Association of MUC13 with inflammation and barrier mediators in the colonic mucosa of IBD patients. (**A**) Relative mRNA expression of *MUC1* and *MUC13* in inflamed colonic biopsies from IBD patients (N = 26; UC = 15, CD = 11) and non-inflamed colonic biopsies from non-IBD controls (N = 10). Significant differences compared to control group are indicated by * *p* < 0.05 (one-way ANOVA). (**B**) Immunohistochemical analysis of MUC1 and MUC13 expression in inflamed colonic biopsies from IBD patients and non-inflamed biopsies from non-IBD controls. Representative images were selected. Scale bars are 20 μm. (**C**) Heatmap visualization of the log2 fold change expression of mucins (*MUC1*, *MUC13*), tight junctions (*CLDN1*, *CLDN2*, *CLDN3*, *CLDN4*, *CLDN7*, *TJP2*), adherens junctions (*CDH1*), cell polarity subunits (*CRB3*, *PRKCZ*, *SCRIB*), potential regulators of intestinal barrier function (*JAK1*, *MAPK1*, *MAPK9*, *ROCK2*, *SNAI1*, *ZEB1*) and regulators of epithelial immunity (*CEACAM1*, *CEACAM5*, *CEACAM6*, *ACE2*, *TLR4*, *IL10RB*, *IL22RA1/2*, *IL-22*) in the colon of IBD patients (N = 14 [7 CD and 7 UC]) and non-IBD controls (N = 4). Significant differences compared to control group are indicated by * *p* < 0.05; ** *p* < 0.01; *** *p* < 0.001. Significantly altered genes that were identified by qPCR are indicated with brackets. # *p* < 0.05. (**D**) Hierarchical clustering and heatmap visualization of the normalized counts of *MUC1*, *MUC13* and other intestinal barrier regulators at individual patient level (IBD [N = 14; 7CD and 7UC]; control [C; N = 4]).

**Figure 9 cells-12-01224-f009:**
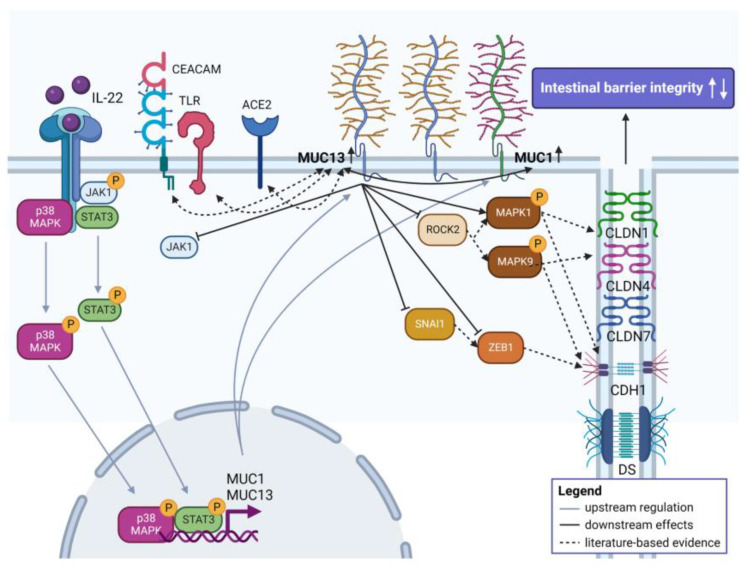
Schematic representation of the mechanism(s) by which aberrant MUC13 expression could affect intestinal barrier function upon IL-22 stimulation. Upon stimulation of IL-22, MUC13 and MUC1 are strongly upregulated via JAK1/STAT and p38 MAPK (only MUC13) signaling. Their increased presence in the intestinal mucosa will impact on intestinal barrier integrity by mediating the SNAI1/ZEB1 and ROCK2/MAPK signaling axes. Figure was created with biorender.com (accessed on 9 December 2022).

## Data Availability

The data presented in this study are available on request from the corresponding author.

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
