# Peer review of "IL-22-Activated *MUC13* Impacts on Colonic Barrier Function through JAK1/STAT3, SNAI1/ZEB1 and ROCK2/MAPK Signaling"

_cells, 2023, doi:10.3390/cells12091224_

Round 1
Reviewer 1 Report
The paper of Breugelmans et al demonstrated an active role for aberrant MUC13 signaling inducing intestinal barrier dysfunction upon inflammation with MUC1 as collaborating partner. This role was demonstrated in both in vitro and in vivo experiments, as well as on IBD patients. The paper is well-written, with an up-to date introduction, adequate materials and methods; the obtained results supported the initial hypothesis. My recommendation is: accept.

Author Response
We thank the reviewer for the overall positive evaluation of our manuscript.
Reviewer 2 Report
The manuscript includes a wide survey of markers of a gut cell line and mice made MUC13 deficient or knockout, respectively, recording some differences, and some that while reaching statistical significance, are biologically unconvincing as the differences are exceedingly small but measured with high precision (resulting in very small error, see point 12). There is at least one example of an inconsistency in the outcomes of an experiment shown in different figures (see point 6). In the final analyses of IBD biopsies, the lack of predictable changes reported in the patient samples confuse the picture but this may not be surprising since the gene knockout is not limited to the intestinal epithelium, nor are most of the measures being made.
I believe the manuscript would benefit from some “streamlining”, bringing focus onto the important data that is consequential to epithelial permeability, and predictions made regarding whether increases or decreases likely contribute to improving or compromising permeability. This is particularly true of the IBD sample analyses. Considering what you discovered regarding changes associated with reduced MUC13, make a prediction on what you might find in the human samples. In fact, this is not easy since most of the story is built on a model in which MUC13 is purposely reduced in cells while the biopsy staining shows MUC13 is increased. Without the predictions, it is very difficult to appreciate how these are related.
Specific criticisms
1. Figure 2, the mRNA and protein data are derived from 24hr stimulations with IL-22, therefore any change is at risk of being irrelevant to IL-22 directly acting on the cells. At 24hrs you cannot conclude either phosphorylated signaling molecule is due directly to IL-22. The pSTAT3 and pJAK should be examined within minutes of applying the IL-22, otherwise the signal risks being confounded by molecules the cells secreted within 24hrs acting back on the cells. With this thought in mind, if you have mRNA sequence data, consider reporting the secreted mediators in your treated cells, like IL-6, IL-10, TGF-beta, etc.
2. Figure 3, how do you manage to get error bars on the “untreated Ctrl siRNA” bars, and some bars do not average to 1, if the signal all other treatments are “normalized” against this? Every repeat yields “1”, and the mean of the repeats is “1” with no error. In Figure 4 I am presuming “ctrl siRNA” is divided by “untreated” (e.g. normalized by dividing the signal by untreated control, as are all other treatment groups. In this case I can understand that the control siRNA groups are very similar to untreated and therefore there a error bars around the average of “1”.
3. Figure 3I is not what the legend describes it to be.
4. line 279, “and Analysis of Variance (ANOVA) tests.” What post hoc tests were done? Few of the data points in Figure 5A were significantly different that I am concerned that the 2way ANOVA test may have returned a non-significant result for Figure 5A, and that “determined to be different using Two-Way ANOVA\” was an incorrect application of the test. The two-way ANOVA is an Omnibus test statistic, it does not provide comparisons between data on a single day (e.g. point in time). It does provide a p value for the overall comparison that is the signal for whether you can proceed to even test between treatment groups on the same day.
5. All Figures, why are you showing nonsignificant differences with horizontal bars for some of the comparisons? If the statistics indicate the comparison does not reach your designated level of significance then they are not different. There is no argument that p values above 0.05 indicates a “near significance” or “trend”. I recommend you read doi: 10.1136/bmj.g2215
6. As I understand the information, Figure 4 fails to reproduce any effect of MUC 13 downregulation on mRNA levels for the listed molecules. The protein and phosphoproteins levels may change, but not mRNA, so I do not understand how this can represent results similar to Figure 1D?
7. Figure 4, I fail to see how the bars in F and H can be derived from the “representative” western blot signals shown in I. Levels of ROCK2 in cells treated with both siRNAs with or with IL-22 appear higher than any other treatment, even taking actin levels into consideration? The western shows minimal difference but the histograms show much more generous differences?
8. The data demonstrating the efficacy of MUC13 knockout in the mice should be included in Figure 5, so the reader can make their own interpretation without consulting the suppl data. MUC13 expression is not limited to the colon, are these mice comparable in weight and growth comparable to WT mice? And for the DSS experiments, were the KO mice co-housed with WT in order to control for gut bacteria?
9. Figure 5A (disease activity) and 5B (endoscopy) are ordinal data and should not have mathematical functions applied, and nonparametric statistics should be used. If TNF is different between the KO and WT mice only at the first DSS cycle, then really none of the cytokines measured are responsible for the heightened colitis and disease activity seen in KO mice. This could be due to MUC13 deficiency in an organ other than the colon, like the esophagus (resulting in eating difficulties, for example). You need to label Figure 5B in a manner that allows better comparison with D – G, because it certainly seems that none of the D- G measure explain differences in disease activity (A) or endoscopy (B), but it is difficult to appreciate when the mice were killed in D- G.
10. Why were Occludin and zonulin not examined? Also, with respect to tight junction proteins, immunofluorescent detection of the molecules in situ provide a much more informative and compelling description of the fate of tight junctions than protein and/or mRNA levels. Proteins may increase or decrease in absolute amount but some of these proteins are not necessarily limited to the tight junctions but may be added to removed despite changes in the amount. The authors admit on line 538, “in addition to its expression in the cytoplasm of colonic enterocytes”, which directly weakens the idea that protein levels alone are important.
11. Figure 6, again, does not show much of the secretory profile of cells in the samples. Moreover, because 6B is not prepared from epithelial cells only, all these signals have to be interpreted with the understanding that mRNAs from the cell infiltrate combined with epithelial cell loss due to ulcers dilute the epithelial cell mRNAs. I will suggest that you “normalize” your mRNAs presumed to be epithelial cell specific to an epithelial cell specific mRNA marker, see for an example: doi: 10.1016/j.ab.2009.12.029
12. Figure 6, interesting that there is very little pathology in the images in C, although this is compatible with the DAI scores reaching only 7 on a scale of 12. The differences, though reaching significance, are truly small when you consider the subjective nature of the data collection and the fact that a parametric statistic was used (mean with standard errors). And your scale includes weight loss which many authors show separately. DSS colitis results in submucosal edema, submucosal and mucosal infiltrate, ulceration, very little of which is clearly visible in these images (maybe a hint of an ulcer on the left side of the MUC1/WT/DSS cycle 3 image). Leaves me concerned with where the samples were taken from the colon. The most effective way to view the DSS inflamed colon is to prepare a Swiss roll, so most, if not the entire colon is observable. The injury is mid- to distal colon. It is not clear where the tissues shown in Figure 6 were collected from.
13. ln 528 “Having identified that JAK/STAT induced-MUC13 signaling could affect intestinal epithelial barrier function via SNAI1/ZEB1 and ROCK2/MAPK signaling pathways upon inflammation,” is an overstatement. You have identified an association, not a mechanistic relationship. There are multiple events that could occur in between knockdown of MUC13 and the changes you register, especially considering the length of exposure to MUC 13siRNA (3 days line 330 up to 10 days for Transwell permeability experiments. In fact, the period of exposure to the siRNA should be more clear in each experiment. Figure 1A show the extent of mRNA depletion only at 72 hours but this is not the only time period used in the study.)
14. In Figure 7D, as the authors state, there are differences which seem to distinguish UC from CD, for example IL-22 and IL-22RA1 (in fact most of the genes listed). Yet levels MUC levels were similar and increased in both UC and CD; how do you reconcile this divergent pattern with everything else you have shown using mice and the cell line? There seems to be no common pattern of events.
minor details
Figure 1B, it is unclear what is meant by “MUC1 untreated control” and “MUC13 untreated control”. I suppose “MUC1 untreated control” means the cells were treated with MUC1 siRNA and nothing else, but it would be less confusing if the label said so. Consider using “MUC1 siRNA only”. Same for C where the labels change to “untreated MUC1”, etc.
The methods say “housekeeping genes Actb and Rpl4 (for the murine samples) and 226 ACTB and GAPDH (for the human samples)”; can you confirm that the housekeeping gene mRNAs did not change with any of the treatments? It is not clear how the housekeeping mRNA levels were applied in this Figure 1.
I recommend re-ordering the GeneClass colours in the Figure 1D in the order that they appear in the figure- makes it easier to follow.
line 199, whether the patients had any drug treatments and the location of their Crohn’s needs to be declared; “intestinal biopsies” is too little information for a disease that can be patchy and small intestinal versus colon- were only colonic biopsies taken?
line 551, “in” is repeated
TNF is no longer called TNF-α, see doi: 10.1001/jamadermatol.2015.4322
line 505 unless you show that MAPK9 is unique to epithelial cells, the upregulation of MAPK9 in MUC13 ko mice may not be limited to epithelial cells
